# Ecological plasticity governs ecosystem services in multilayer networks

Clare Gray[1,2], Athen Ma[3], Orla McLaughlin[4], Sandrine Petit[4], Guy Woodward [2] & David A. Bohan [4✉]

Agriculture is under pressure to achieve sustainable development goals for biodiversity and ecosystem services. Services in agro-ecosystems are typically driven by key species, and changes in the community composition and species abundance can have multifaceted effects. Assessment of individual services overlooks co-variance between different, but related, services coupled by a common group of species. This partial view ignores how effects propagate through an ecosystem. We conduct an analysis of 374 agricultural multilayer networks of two related services of weed seed regulation and gastropod mollusc predation delivered by carabid beetles. We found that weed seed regulation increased with the herbivore predation interaction frequency, computed from the network of trophic links between carabids and weed seeds in the herbivore layer. Weed seed regulation and herbivore interaction frequencies declined as the interaction frequencies between carabids and molluscs in the carnivore layer increased. This suggests that carabids can switch to gastropod predation with community change, and that link turnover rewires the herbivore and carnivore network layers affecting seed regulation. Our study reveals that ecosystem services are governed by ecological plasticity in structurally complex, multi-layer networks. Sustainable management therefore needs to go beyond the autecological approaches to ecosystem services that predominate, particularly in agriculture.

[1] Queen Mary University of London, School of Biological and Chemical Sciences, Mile End Road, London E1 4NS, UK. [2] Department of Life Sciences, Silwood Park Campus, Imperial College London, Ascot, Berkshire SL5 7PY, UK. [3] Queen Mary University of London, School of Electronic Engineering and Computer Science, Mile End Road, London E1 4NS, UK. [4] Agroécologie, AgroSup Dijon, INRAe, Univ. Bourgogne, Univ. Bourgogne Franche-Comté, F-21000 Dijon, France. ✉email: david.bohan@inrae.fr

A basic tenet of network ecology is that functions and processes arise from the underlying structure of interaction networks. Ecosystem functions can be delivered either by an individual species or by many, and these contributions are shaped by their interactions, both directly and indirectly. Consequently, management directed at network structure, supporting not only the presence and abundance of key species nodes but also their interactions, should give the best chance of maintaining ecosystem functioning[1–4]. Ecosystem services (ES) are ecological functions that have a social and/or economic value that we wish to maintain and manage[5]. Network structural governance of ES and the interplay between different ESs has rarely been demonstrated empirically and little is known about how best to manage network structure to support services, largely because very few datasets that quantify ES in a network context have been assembled (but see Pocock et al. [6] for a notable exception).

In what has come to be termed multilayer networks[7–10], where ecological networks are clustered into separate but interlinked network layers by their functional properties, the complex structuring of species nodes, links and node switching across different network layers is hypothesised to govern the delivery, stability and balance of functions in an ecological community[6,11]. Within this framework, key predator species that contribute to the service of weed-seed regulation by feeding on weed seeds in one layer, might also disrupt this function by switching to the consumption of alternative prey species[12,13], such as gastropods, in another layer where the amount of alternative prey increases. In switching between prey, and therefore functions, the predators move between the layers of the full ecological network. Understanding these interlayer dynamics will provide new insights into the direct and indirect effects of biodiversity loss within an ecosystem and help devise more effective management for maintaining ES.

Our focal ES multilayer network is based on the group of key carabid beetle species and their interactions in arable agroecosystems. Communities of carabid beetles typically contain a mix of species from specialist granivores to more generalist predators, each with differing capacities to switch to alternative prey and between network layers. Classical ecological approaches have demonstrated that weed seedbank change is related to the abundance of particular groups of carabid species. These species prey upon weed seeds at the soil surface and reduce the amount of viable seed that can enter the seedbank, thus regulating the weed population[14,15]. Molecular trophic analysis of carabid gut contents has revealed that gastropods are important prey[16] and that carabid predation can modify their field-scale spatio-temporal distributions[17]. Typically, such services of weed seed regulation and pest predation are studied autecologically, in isolation. A network analysis of a joined-up weed seed regulation and pest predation system could have been done using tripartite[18,19] or coupled network[20] approaches. The multilayer network approach, as we use it, allows us to consider weed seed regulation and pest predation as two otherwise simple trophic networks, the layers, whose interaction can also be analysed by the consideration of the carabid nodes that are shared. Our practical interest is to understand whether changes in weed seed regulation across the networks are best explained either by the carabid species present switching between alternative prey[21,22], and thereby forming new networks links (link turnover), or by change in the composition of carabid species present (species turnover) or by some combination of the two.

We analyse the interlayer dynamics between weed and pest ES using data from the Farm Scale Evaluations (FSE) of genetically modified, herbicide-tolerant crops in which ecosystem functions were measured at multiple sites at the Great Britain, national scale[23] ("Methods"). The FSEs evaluated the effects of herbicide management of conventional and Genetically Modified (GM) crops on farmland biodiversity in 187 fields in a split-field design (374 half-fields) of three spring-sown crops. Different herbicide management was applied across the conventional and GM treatments, and between fields and crops, and this perturbation produced distinct community combinations of species of weed seeds, gastropods and carabids in each split field. The nodes of the 374 two-layer networks we analyse are formed from the species of weeds, gastropods and carabids in the dataset, and trophic interactions collated from the literature (Methods). First, we tested the hypothesis that the ES of regulation of the weed seedbank is determined by the carabid-weed seed, herbivore layer. We then tested whether weed seedbank regulation co-varies with pest gastropod mollusc predation, by including the trophic interactions between the carabids and the gastropod molluscs in the carnivore layer. Finally, we tested whether change in the ES of weed seedbank regulation and structure of the network layers was explained, or governed, by link turnover that rewires the network between the herbivore and carnivore layers, via prey switching. Our results reveal that carabids can switch to gastropod predation with community change, and that link turnover rewires the herbivore and carnivore network layers affecting weed seed regulation. This suggests that ESs are governed by ecological plasticity in structurally complex, multilayer networks.

## Results and discussion

The composite network, formed by the fusion of all 374 ecological networks, consisted of 811 hypothesised, pairwise trophic links between 41 carabid, 96 weed and 9 gastropod species (Fig. 1). Seventeen species of carabids were obligate herbivores, with links only to the weed seeds; six were carnivores, feeding only on gastropods. Eighteen were omnivorous species, capable of switching their diet depending upon the context and thereby switching between being herbivores or carnivores. We found no pattern in the richness of weed and gastropod species (Supplementary Fig. S1a).

In the herbivore layers, we found that seedbank regulation increased with carabid species richness for the total ($F_{1,328} = 4.9$, $p = 0.0275$, Supplementary Fig. S2a), monocotyledon ($F_{1,328} = 6.4$, $p = 0.0117$, Supplementary Fig. S2b) and dicotyledon ($F_{1,328} = 5.1$, $p = 0.0246$, Supplementary Fig. S2c) components of the weed community. We found that the greater the summed predation interaction frequencies of carabid—weed seed herbivore links, the greater the regulation of the total weed ($F_{1,328} = 5.2$, $p = 0.023$, Fig. 2a), monocotyledon ($F_{1,328} = 3.9$, $p = 0.049$, Fig. 2b) and dicotyledon ($F_{1,328} = 5.7$, $p = 0.017$, Fig. 2c) components of the seedbank. In summary, the regulation of the weed seedbank in each split-field was altered by both the count and frequency of herbivore links among carabids, confirming our first hypotheses for the driving role of the herbivore layer on weed seed regulation.

We collated the herbivore layers with the carnivore layers and found that the species richness of carabids acting as herbivores in the herbivore layers decreased as the richness of gastropod species increased in the carnivore layers ($F_{1,367} = 182.1$, $p \leq 0.001$, Supplementary Table S1 and Supplementary Fig. S1b). The number of hypothesised links from carabids to the weed seeds and to the gastropods in the layers of each network also co-varied inversely ($F_{1,3672} = 212.8$, $p \leq 0.001$, Supplementary Fig. S1c). The calculated herbivore layer predation interaction frequency decreased inversely and dramatically as the predation interaction frequencies of the carnivore layer increased (Fig. 3a). This co-variation could represent a trade-off between the number of links to weed resources and number of links to gastropod resources, as no carabid node behaving as an omnivore was highly linked to

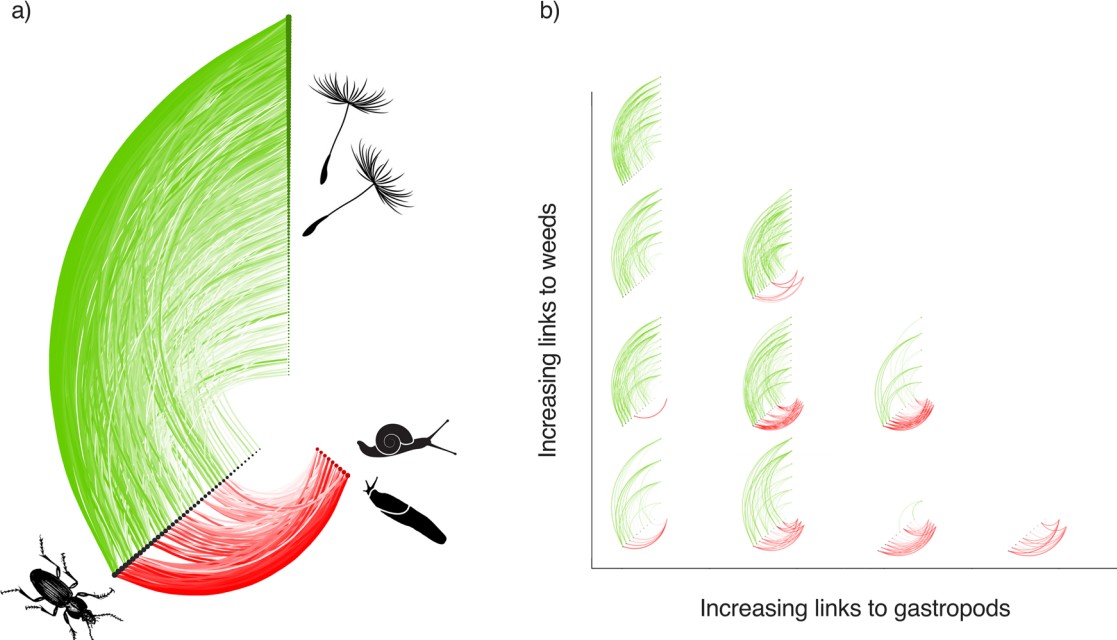

**Fig. 1 Hive plots of carabid, weed and gastropod mollusc networks. a** The 'composite network' encompassing all species and their interactions across all the multilayer networks from all sites used in this study. Carabid (black circles), weed (green circles) and gastropod mollusc (red circles) species nodes are sized proportionally to their rank of how often they were found across all networks. Link colour indicates the herbivore (green) and carnivore (red) layers, with the colour intensity and thickness being proportional to the strength of the interaction across all networks. **b** Example networks selected to show the trade-off between the number of links to the weeds and to the gastropods. The images of the beetle, weed seeds and molluscs are reproduced under the standard royalty free licence of 123rf.com (vectorgalaxy © 123RF.com; Patrick Guenette © 123RF.com; yod67 © 123RF.com; Autt Khamkhaunchan © 123RF.com).

both weeds and gastropods in any network ($t = -7.08$, $p = <0.001$, Supplementary Table S1 and Fig. 3b). This pattern across the networks suggests that weed seedbank regulation co-varied in inverse relation to pest gastropod mollusc predation via changes in the structure of the carnivore layer, supporting our second hypothesis.

Species turnover in carabid composition across all the networks, measured as mean dissimilarity, was $0.44 \pm 0.17$ (Supplementary Fig. S3), and many of the same carabid species were present in the majority of networks. The structural changes in the networks were more related to the turnover of links between the carabids and their resources (mean dissimilarity $0.65 \pm 0.25$, Fig. 3c) than to changes in carabid species composition per se. This supports our third hypothesis that link turnover of the network layers governs the delivery of weed seed regulation. Such link plasticity can be visualised as occupancy of the total potential link space, by each carabid species (Fig. 1b). For instance, the common carabid species in the dataset, *Pterostichus melanarius*, previously associated with in-field mollusc and weed seedbank regulation[14,17], operates here as a specialist carnivore or a herbivore or an omnivore in different networks depending on the conditions of weed and gastropod resources (Fig. 4). Trophic link plasticity was typical of a number of the most abundant carabid species, such as *P. madidus*, *P. niger*, *Harpalus rufipes* and *H. affinis*, in the networks (Supplementary Fig. S4) potentially marking them out as key species that govern ES delivery. Higher dissimilarities at predator link values of 5 and above may reflect carabid predation preferences for molluscs, which remain to be tested.

This study explicitly relates the variation in structure across replicated multilayer ecological networks to associated ESs. Changes in the abundance and/or presence of some species of alternative prey are expected to lead to marked turnover in

trophic link structure[24]. Here we found that network structural change was consistent with carabid species switching between weed seed and gastropod food items, with many carabid species in the network being behaviourally plastic and playing roles of carnivore, herbivore or omnivore in different manifestations of the network with differing community contexts of weed and gastropod nodes. The considerable variation in structure observed across all network examples, in this composite network from the same ecosystem, was consistent with network rewiring[25,26].

The site-specific multilayer networks analysed here are subsets of nodes and links, drawn from a larger, composite network of interactions across all half-fields in the dataset (Fig. 1a, b). As such, the networks are dependent upon the quality of the trophic interaction information currently in the literature. There were many prey species present in the dataset for which no trophic information could be found and which were consequently excluded from the networks and network analysis. The species excluded were predominantly weed species (109 species removed), but also some gastropod species (3 species removed). Links were also assigned to carabids at the genus level. These processes could lead both to an under- and an over-estimate of predation on some prey. While the current state of knowledge of the trophic links between all species present in arable farmland is far from complete, it is being resolved through molecular analysis of gut contents of carabids and other consumer groups[22,27–29]. We expect that the resulting higher resolution networks will confirm the results we present here.

It is now being recognised that carabid species have far more versatility in their diet than previously thought[27] and that diet range might change with community diversity, as a function of behaviour, resource availability and food choice[30]. Our results highlight how the plastic, synecological traits of species, as opposed to their fixed, autecological traits, shape their ultimate

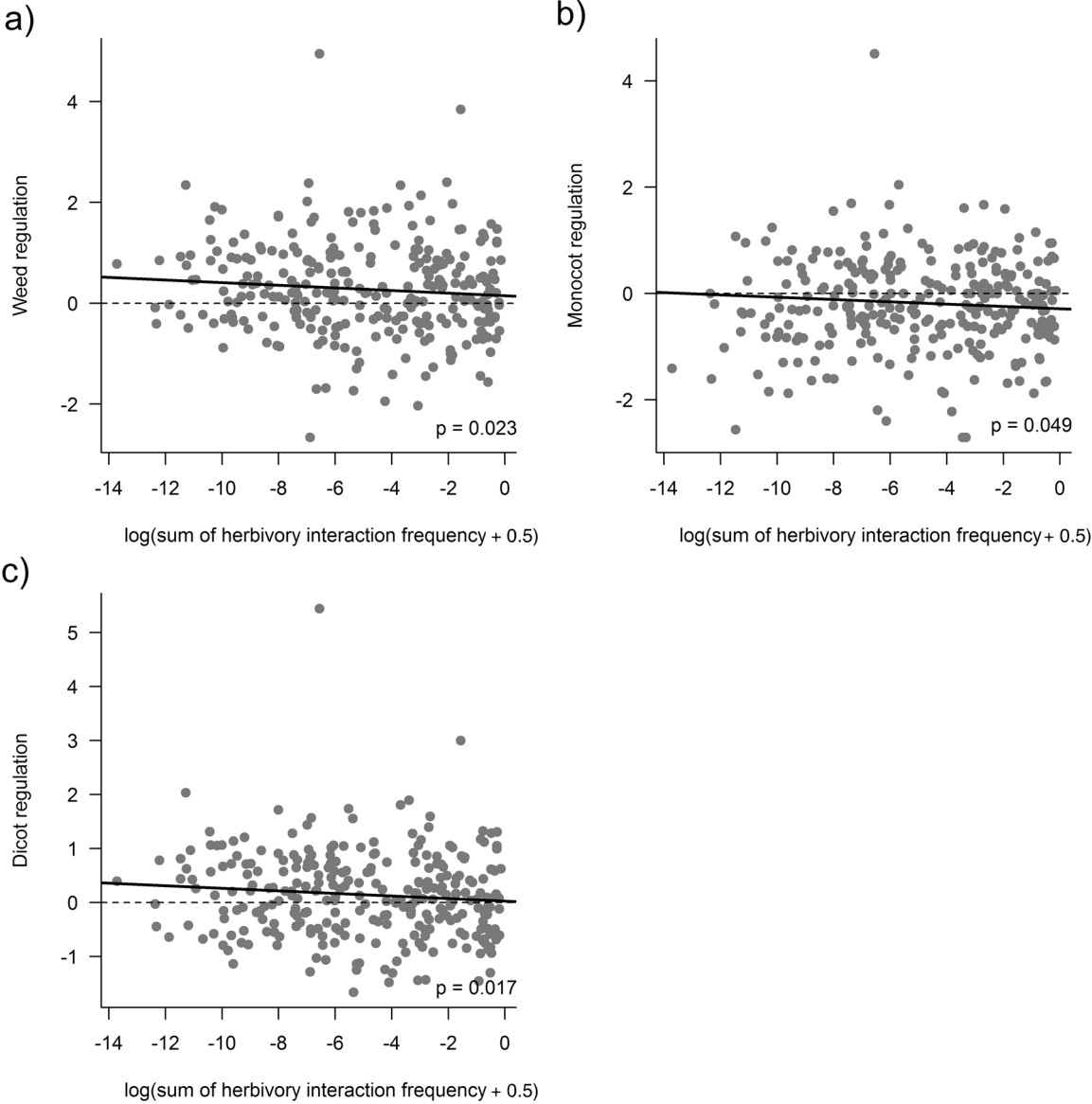

**Fig. 2 Weed seed regulation increases with herbivory interaction frequency.** The level of weed regulation in relation to the log of the summed herbivory predation interaction frequency in each network for: (**a**) total weed regulation; (**b**) monocotyledon weed regulation; and, (**c**) dicotyledon weed regulation. Solid lines in (**a**), (**b**) and (**c**) are regressions from GLMMs (Methods, Table S1). The dotted line at 0 indicates the threshold where weed seedbanks decline between $t_0$ and $t_1$.

contribution toward ecosystem functioning. Ecosystem management, targeted at the delivery of desired services, should consider the variation in interactions and food choices underpinning those services rather than simplistic trophic, autecological roles. Abundant carabids, such as *P. melanarius*, have important roles in weed seed predation and regulation[14,22,29] and are highly flexible consuming both weed seeds and gastropods in different amounts depending on the context of the network in which they are embedded. Directing management towards the weed seed consumers alone, will not deliver stable service provision[14], due to the trophic plasticity many of these species demonstrate (Fig. 4 and Supplementary Fig. S4). Rather, management of both species nodes and links that increase the specialist carabid-weed seed interactions could assure more stable service delivery. For example, future attempts at the reduction of herbicides in agriculture by employing carabid weed seed regulation might also be joined to efforts at gastropod control using molluscicides within novel Integrated Weed and Pest Management systems. This

mechanism, and whether it would result or not in pesticide reduction, could be tested experimentally by manipulating molluscicides and herbicides applications to model communities of carabids, weeds and gastropods, with molecular biological characterisation of the resulting carabid diets[27,28].

## Methods

**Experimental design and data collection**. The data came from a highly replicated field trial (The Farm Scale Evaluations, FSE) designed to test whether the adoption of genetically modified, herbicide-tolerant (GMHT) crops would have led to significant changes in UK farmland biodiversity by comparison with the conventional crops and herbicide management then used[23]. The FSE sampled the biodiversity in and around 187 spring sown crop fields across the UK using a standard methodology (Supplementary Fig. S5). Post-hoc analyses demonstrated that the trial design had more than sufficient statistical power to test the null hypothesis of no change in biodiversity[31] and indicated that the species nodes were fully sampled[32,33]. Full details of the experimental design and protocols for data collection can be found in Champion et al.[34] and Bohan et al.[35], which we briefly detail here.

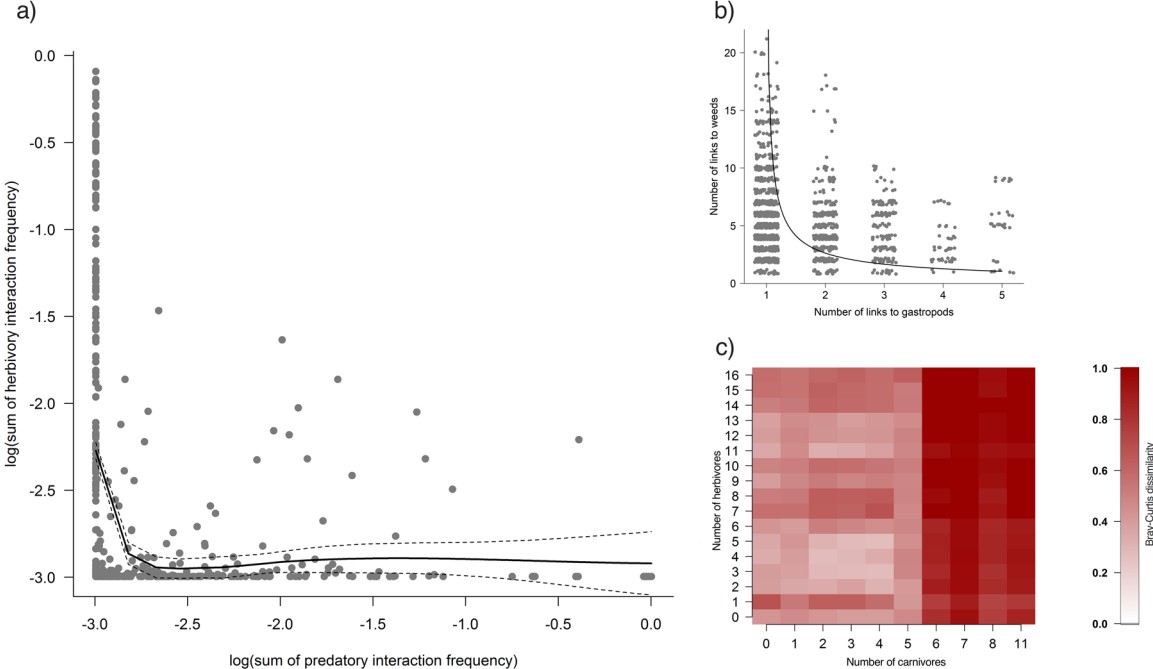

**Fig. 3 Trade-offs between herbivory and carnivory links. a** The relationship between the sum of specialist herbivory frequency interaction frequency and sum of specialist carnivory interaction frequency, transformed to the log(x + 0.5) scale, for each network. The line shown represents a LOESS smoother with dashed standard errors. **b** The number of links to weeds and gastropods for the omnivore carabid nodes within each network. The line shown is a regression fitted using a GLMM ("Methods", Table S1). **c** The estimated link turnover between networks across the herbivore/carnivore gradient using Bray–Curtis dissimilarity.

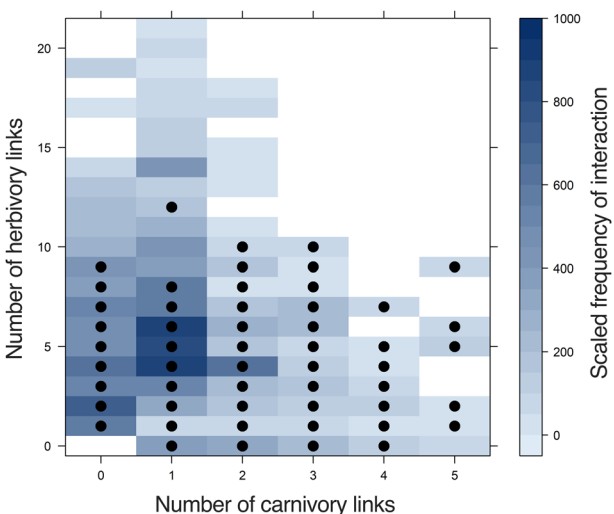

**Fig. 4 Link plasticity among carabid species.** Density plot showing the relationship between the number of herbivore and carnivore links across all networks. Each blue-shaded square indicates the instances that an omnivore carabid had a particular number of links to gastropods and to weeds, with the depth of the blue shading indicating the frequency of observation across all networks, scaled between 0 and 1000. Some networks contained carabids acting as pure herbivores or carnivores in the networks, but most contained omnivores. A filled, black circle is used to denote the link composition observed in different networks for the most common carabid species, *Pterostichus melanarius*, which in some networks appeared as either an herbivore (having links only to weeds) or a carnivore (having links only to gastropods), but more often acted as a flexible omnivore.

The count data for the carabids, weed seedbank and seed rain, and gastropod molluscs came from 66 spring-sown beet, 55 spring maize and 66 spring oilseed rape fields. The fields were distributed across the UK (Supplementary Fig. S5) and each field was sampled for one cropping year[23] between 2000 and 2004. The great majority of fields were below 15 Ha in size, and typically varied between 2 and 10 Ha depending upon the crop[34,35]. The trial used a half-field design, and each field was divided in two with one half sown with a conventional crop and the other a GMHT variety of the same crop. Data from both treatments are used for the analyses presented in this study, giving a total of 374 half-fields.

The pitfall trapping of soil-surface-active invertebrates employed the method described by Brooks et al. [36]. Pitfall traps, spaced at 2, 8 and 32 m points along 4 transects running from the field-edge into the cropped area, were opened in the spring (April / May) and summer (June / July) and in late summer (August). The weed seedbank was measured by taking 8 soil samples in each field, at 2 and 32 m points on four transects, prior to the crops being sown and after harvest[37]. The soil samples in each half-field were bulked, placed in germination trays in a greenhouse and the seeds they contained allowed to germinate over the course of 18 weeks[37,38]. The viable weed seeds available to the carabids were measured as the seed rain onto the soil surface from the weed plants in the field. Eight seed rain traps, placed at 2 and 32 m along 4 of the transects, sampled the rain of weed seed throughout the growing season[37]. Gastropods were sampled using baited refuge traps at the same positions used for the pitfall trapping in late April and in early August for spring oilseed rape, and in May and August for maize and beet[36]. All carabids, molluscs and weed seeds sampled were identified as species and counted. For the carabids and molluscs, counts were then pooled, by summation, to give a year-total estimate for each species in each half-field, from which a relative abundance of each species was calculated by dividing the count of that species by the total count for the group of carabids and molluscs, respectively. Total, monocotyledon and dicotyledon weed species counts were pooled, by summation, to give an estimate of these weed seedbanks before ($t_0$) and after ($t_1$) the crop was sown and harvested, respectively.

**Network construction.** The species sample data were supplemented with carabid dietary information recovered from the literature. We assumed that where a carabid species, *A*, was noted to consume a resource species, *B*, in the literature and both these species were present in a half-field, then this trophic interaction was realised[6,39–41]. To standardise the (trophic interaction) sampling effort across all carabid species and reflect the generalist nature of carabid consumers it was assumed, following Honek et al. [42], that each carabid species would consume the same resources as other carabid species within the same genus[39,41]. A similar

generalisation was made at the resource level: where a particular species of carabid was recorded to feed upon one species of gastropod or weed in the literature, we assumed that this carabid would also consume other resource species of the same genus[43]. This assumption of generalisation was done to reduce the numbers of artefactually isolated species within each network, and to avoid the bias towards more highly-studied species[44,45].

The interaction frequency for each realised link between consumer and resource was calculated as the multiplicative product of the consumer and resource relative abundances, under the assumption that species abundance is a predictor of the strength of interaction between species[46]. Applying a frequency weighting to the links in this manner integrates a quantitative estimate of the interaction strength between species in each network, enriching the simple binary network structure built from presence/absence data. Future consideration of carabid interaction strength prey may include traits, such as body size and mandible type[47]. Following construction of the carabid-weed seed (herbivore) and carabid-gastropod (carnivore) layers, each carabid species was assigned to an empirical trophic group based upon their role in each replicate multilayer network. Carabid nodes linked only to gastropods were assigned to the 'carnivore' grouping, while those consuming only weeds were 'herbivores', and 'omnivores' were generalist species linked to both gastropods and weeds. Thus, a particular carabid species might be a carnivore, an herbivore or an omnivore in different replicate multilayer networks. Some species may therefore appear in different classifications across the collection of networks; a particular carabid species may be a weed consuming herbivore in some networks, a gastropod carnivore in others and an omnivore in yet other networks.

**Regulation of the weeds in the seedbank**. Regulation was calculated from the change in total, monocotyledon and dicotyledon seedbank counts between $t_0$ and $t_1$, as:

$$\text{regulation} = \ln\left(\frac{t_1 + 0.5}{t_0 + 0.5}\right) \qquad (1)$$

so that for each half-field three metrics of regulation for the total, dicotyledon and monocotyledon seedbanks were calculated. Negative values of this metric indicate a decline in the size of the weed seedbank from $t_0$ to $t_1$. Following Bohan et al. [14], negative relationships between the metric of regulation and carabid counts are indicative of seedbank regulation by these beetles.

**Statistics and reproducibility**. All analysis was done in R (R Core Team 2019) using the cheddar[48], bipartite[49] and vegan[50] packages. Network plots were created with the HiveR package[51]. Interaction frequencies were calculated separately for herbivores, carnivores and omnivores in each network.

Species and link turnover across the collection of multilayer networks were measured using Bray-Curtis dissimilarity in the vegan package[50]. Each network layer is a realisation of the interactions drawn from the composite network[52], with the interactions only being contingent on local species composition and abundances. To assess how species and link turnover changes across the herbivore/carnivore gradient, we used the number of herbivores and carnivores within each network as factor levels with which to categorise the networks (i.e. networks with 1 herbivore, 2 herbivores, 3 herbivores, etc.). We ensured that no single network appeared in more than one group by randomly assigning networks to either their herbivore or carnivore grouping, and then calculated the Bray-Curtis dissimilarity between the herbivore and carnivore groups.

The regression modelling between species and link variables was done using Generalised Linear Mixed-effects Models (GLMM) with Gaussian errors. Each field was treated as a replicate as there was no repeat sampling from any one site[23]. All analyses were appropriately subjected to tests of normality and behaviour of the model residuals[53].

*Species richness*. GLMMs were used to assess the effect that increasing numbers of gastropod species has on the number of weed species, and the number of herbivorous carabids implicated in each multilayer network. Field identity was nested within crop type (spring-sown beet, spring maize or spring oilseed rape), which was nested within management type (conventional or GMHT) and included as a random effect. The species richness of all groups was $\log(x+0.5)$ transformed to conform to normality. The relationship between the number of carabid species acting as herbivores and carnivores in each network was assessed using the same model structure.

*Link richness*. The relationship between the number of links to plant resources and links to gastropod resources for the omnivorous carabid nodes was assessed using GLMMs, with field identity nested within crop type (spring-sown beet, spring maize or spring oilseed rape), which was nested within management type (conventional or GMHT) and included as a random effect.

*Link frequency*. Due to the extreme distribution pattern of the carnivore and herbivore predation interaction frequencies, the correlation between these two variables was examined on the $\log(x + 0.5)$ scale using LOESS smoothing[53].

*Weed seed regulation*. The relationship between weed seed regulation and the count of herbivores or herbivory interaction frequency was modelled with GLMMs. The count of herbivores and herbivore interaction frequency were $\log(x+0.5)$ transformed to attain normality. Field identity was nested within crop type (spring-sown beet, spring maize or spring oilseed rape), which was nested within management type (conventional or GMHT) and included as a random effect.

**Reporting summary**. Further information on research design is available in the Nature Research Reporting Summary linked to this article.

## Data availability

The raw FSE data are free from intellectual property rights. The data can be requested by enquiry to the Environmental Information Data Centre of the Centre for Ecology and Hydrology (http://eidc.ceh.ac.uk/contact). Archived information about the FSEs are available from the National Archives of The Government of the United Kingdom (http://webarchive.nationalarchives.gov.uk/20080306073937/http://www.defra.gov.uk/environment/gm/fse/).

Network data for the analyses that support the findings of this study are available from the public repository Zenodo[54]: https://doi.org/10.5281/zenodo.4268722. Requests for any further information can be addressed to the corresponding author.

## Code availability

The R code used to analyse the food web communities is available from the public repository Zenodo[54]: https://doi.org/10.5281/zenodo.4268722. Requests for any further information can be addressed to the corresponding author.

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

## Acknowledgements

D.A.B acknowledges the support of the FACCE SURPLUS *PREAR*, ERA-NET C-IPM *BioAWARE* and ANR (ANR-17-CE32-011) *NGB* projects. D.A.B would like to thank Lucile Muneret for her comments on an earlier draft.

## Author contributions

C.G. and D.A.B designed the research. D.A.B contributed materials and datasets. C.G., A.M., O.M., S.P., G.W. and D.A.B. discussed the results. C.G., A.M. and D.A.B. led paper writing with input from all authors.

## Competing interests

The authors declare no competing interests.
