## [Peer Review File · Communications Biology]

REVIEWERS' COMMENTS:

Reviewer #1 (Remarks to the Author):

I enjoyed reading the manuscript "Ecosystem services are governed by ecological plasticity in multilayer networks" by Gray and colleagues, in which they attempt to bring together the fields of ecological multilayer network and ecosystem services. It expands the use of ecological networks to understand the ecological mechanisms that drive the delivery of different ecosystem services. Specifically, it tries to apply the joint analysis of two important ecosystem services, weed seed and gastropod pest control, delivered by carabids through their consumption of seeds and gastropods, and how it may be affected by the distinct composition of interactions networks. This is important as the we clearly more and more aware that management of ecological systems and ecosystem services can not be done in isolation, and it should be taken an approach that brings them altogether to understand their complexity and interplay.

The authors make a commendable use an extensive dataset of 374 interaction networks – the weed seed-carabids-gastropods tripartite network - obtained through from agricultural fields sampled within the Farm Scale Evaluations project, which strengthens the analysis and allows to uncover potential patterns across the networks.

They aim to test: 1) if ES of weed seed is primarily regulated at the herbivore layer (seed-carabid interactions); 2) if the weed seed control co-varies with carabids carnivore layer (predation of gastropod pest by carabids); and, finally, 3) if the seed weed regulation is affected by the turn over in the type of interactions established by carabids, i.e. switch between herbivory and carnivory.

The main conclusions are: 1) that weed control is improved (perhaps not surprisingly) by the increase in the frequency of herbivore interactions, 2) weed control is negatively affected the increase in carnivory by carabids, and 3) community changes leading to distinct availability of resources rewires the interactions by carabids between the herbivory and carnivory. Overall, the authors conclude that the ecological plasticity in the carabids networks has important consequences for ecosystem services and that multiple ecological functions need to be considered for the sustainable management of natural and agricultural systems.

Overall, the manuscript is well written and easy to read, and the authors managed to deliver the message in a clear, and concise way. Procedures for network building and statistical analysis are generally provided in sufficient detail so the reader can follow the steps taken by the authors. Bibliography seems enough to contextualize the work and discussion of the results.

There are a few points however I wish the authors to clarify and are detailed below.

Good work and good luck.

General comments:

1 - The authors put quite some emphasis on the multilayer aspect of ecological functions and ecosystems services and frame their work within that context (L.37-48). Though it is true that their system is naturally multi-layered, and should be mentioned as it is done, one aspect of the networks assembled here is that they actually miss explicit interlayer links connecting both layers. As such, what we have here is a network made up of three groups of species (weed seed – carabids - gastropods) that conform to a tripartite network (e.g. Boulay et al. 2009, Correia et al. 2019) or coupled/interdependent networks (Albrecht et al. 2014) and I think that is the term that should be used to classify the networks in this work (L.49).

As stated below, this work focus on the dynamics of the resource use by the guild that connects both layers of the system, and how that may spread across the two types of interactions (perhaps check Correia et al. 2019 or Lopez-Núñez et al 2017).

2 – The analysis of links and interaction frequencies is largely focused on the global numbers (sum of herbivore links, sum of interaction frequencies, etc), which give a global view of the participation of the carabid guild connecting. However, the authors could go a little further by looking at the variation in the links of each species between layer across all networks. Perhaps considering some simple species-level metrics would eventually be of interest, and could be integrated in the section on herbivory-carnivory relationship (paragraph starting on L.98). For instance, carabid species degree or generality/vulnerability (both depicting the nr of partners, though in different ways. Check bipartite package's document for details and refs) for each interaction type could be obtained to calculate a herbivory:predation ratio that could be then related to the variation in gastropod/plant richness/abundance across all networks, or even with seedbank regulation. Eventually, also looking for the omnivore carabids could check if there's some tendency (e.g. using a degree-degree correlation) for the same species to have the same role across ecological functions.

3 – Regarding the assumptions the authors made in building the interaction networks I think they are sensible, but I would say that inevitably they are not perfect and may have some potential drawbacks. I feel that they deserve some consideration and should be presented and discussed. This can be done either where assumptions are presented or in a section of the discussion. Specifically:

3a - The assumption of diet generalisation by carabids (L.206-208) may over-estimate the existing links. Alternatively, a rarefaction approach could be used, which in turn may under-estimate the links actually occurring in the fields. Perhaps, a brief discussion of the potential effect of the assumption of a wide diet generalisation by the carabids could be added to the discussion.

3b - I agree that applying this frequency weighting to links based on the product of species abundances (L.209-212) it's better than just a simple binary network. However, abundance is not the sole determinant of interaction strength, and other factors such as species preferences may result in deviations from the simple product of abundances (Vázquez et al 2005 or Staniczenko et al. 2013), which could also be affect for example by the connectance of the network. Do you think that the interaction strengths would be much affected if they were not only based on abundances? Perhaps a brief discussion of these assumption should be provided.

4 – Finally, the authors seem to imply the existence of a trade-off in managing the two types of organisms (L.156-158) not desired at crop fields (weeds and gasstropods): to reduce the use of herbicides one should focus on reducing the potential to prey-switching towards gastropods (due to the trophic plasticity by carabids), which may require an extra input of molluscicides. In that case, we are just moving from one chemical to another. Does it mean that it wouldn't be possible to make a joint reduction of the two types of external inputs? Does it suggest that such inputs are inevitable, and we are just faced with a decision about the ration of those inputs?

Detailed comments to the authors:

L.65-66: After Farm Scale Evaluations add "FSE".

L.73: Quite not understand this. The nodes of the network are the actual species and not their abundances. Please, clarify or/and remove "abundances".

Section "Results and discussion": I would suggest dividing this section according to the three hypothesis state above to make it easier to follow. Unless, not allowed by the formatting rules of the journal.

1st paragraph: It would be interesting, as a general overview, to a have a summary of the interactions (total and for each type) and species across all networks (mean and range).

L.83: Aggregated networks usually means that all layers are merged to form a single monolayer network that sums up all the nodes and links of the multilayer network (Kivelä et al 2014, pp. 228), which is not the case here, and it's not synonymous of metaweb. Better remove "aggregated network" here and call it "tripartite metaweb".

L.102: But Fig2c reports the number of herbivores vs number of predators (which could read as number of nodes), rather than the number of links of each type established by carabids. Please clarify.

L.107: I think this plot is worth being as main figure, as it helps showing the general separation between the two functions provided by different species of carabids, and that omnivore species are not that efficient at delivering both functions. Perhaps as an extra panel of fig3.

L.110-114: It can be noticed on both figure 3b and figure S5 an apparent change in turnover once the number of carnivore species reaches a certain number of species (transition from 5 to 6 species). This seems to be ignored, but I wonder if there's some reason/meaning for that.

L.116: The figure on the left could be a nice visualization and I like how it looks, but I don't think it works very well here. For example, the first column of networks (no links to gastropods) should have no red links but they are present. Similar thing happens on the rows. It may be a matter of size or space available. Also, isn't it showing the same as fig S4? Consider removing or redrawing. If figure is kept add a) and b) to the respective panels.

L.141: Close parenthesis missing after references.

L.169: Perhaps add a scheme showing the split-field design used.

L.171-172: I understand that all the details are already published in previous works and there's no need (or the space) to give them again here. However, scale at which the data were collected (plot size), distance between fields, or an idea of how independent the half-fields are should be provided, I think.

L.183: How many soil samples from each field were collected?

L.187-188: How many seed traps?

L.193-194: Given that seedbank total of weed species was obtained as a before-after, does the relative abundance refers only to animals (gastropods and carabids)? If that is so make explicit. If for weeds the relative abundance refers to the before-after number of seeds, perhaps describe it in a separate sentence. Otherwise, seems a bit confusing as it is. Moreover, before and after what? Reckon, as suggested further down in L. 224-225, that it refers to harvest of crops at each field. It is not clear. Also, is this the difference between before and after, or the sum of counts in all samples before and after. If the later, why are you summing the counts from both instances? Clarify, please.

L.202: Individual? Species?

L203-206: Isn't this saying the same as the previous sentence? Seems redundant.

L.224: If I correctly understand, any deviation from unit (i.e. no seed difference between t1 and t0) would be due to carabids regulating the seedbank, right? Isn't this a too strong assumption? This would imply that any other factors (ant predation, wind, or other types of dispersal, and factors) are mostly negligible.

L.240: But it seems that the decomposition of link dissimilarity was not actually done here. In the paper cited at the end of this sentence (Poisot et al 2012) it is proposed an approach to perform that decomposition, and you could present not only species and interaction turnover, but also how much of the interaction turnover is due to changes in species composition and to the same species establishing distinct interactions - rewiring. Perhaps that approach could be employed here as well. In a more detailed approach, interaction turnover due to species turnover could be further decomposed into the contribution prey turnover, carabid turnover and simultaneous turnover of plant and carabids (Novotny 2009, Simanonok & Burkle, 2014, Arroyo-Correa et al. 2020).

L.252, 260, 265, 269: All these models are mixed models, so perhaps write it explicitly. Moreover, all

of them seem to use Gaussian error distribution, given the transformation applied to data, except in the case of the analysis of link richness. In this later case, which error distribution was used? If in all cases Gaussian error distribution, perhaps simply call the models Linear Mixed Models.

Figure 1: I'm not entirely convinced that this particular display of the network is very effective, given the high number of links involved. Perhaps a simpler display, with weeds at the bottom, carabids in the middle and gastropods at the top, spread across the width of the page would allow a better view of the profusion on links in the metaweb.

Figure 2: Add to the legend what is the meaning of the lines, and how to interpret the dashed line. Same for Figure S3 and S4.

Figure 3: Add next to the colour scale on the right that it refers to dissimilarity gradient. Same for Figure S5.

Figure S2: Perhaps organize the different plot as in the Figure S3.

Suggested bibliography (cite, or not, at your discretion):

Boulay R, Carro F, Soriguer RC, Cerdá X. 2009 Small-scale indirect effects determine the outcome of a tripartite plant-disperser-granivore interaction. *Oecologia* 161, 529–37. (doi:10.1007/s00442-009-1404-z)

Correia M, Rodríguez-Echeverría S, Timóteo S, Freitas H, Heleno R. 2019 Integrating plant species contribution to mycorrhizal and seed dispersal mutualistic networks. *Biol. Lett.* 15, 20180770. (doi:10.1098/rsbl.2018.0770)

Albrecht J, Gertrud Berens D, Jaroszewicz B, Selva N, Brandl R, Farwig N. 2014 Correlated loss of ecosystem services in coupled mutualistic networks. *Nat. Commun.* 5, 3810. (doi:10.1038/ncomms4810)

Kivelä M, Arenas A, Barthelemy M, Gleeson JP, Moreno Y, Porter MA. 2014 Multilayer networks. *J. Complex Networks* 2, 203–271. (doi:10.1093/comnet/cnu016)

Vázquez DP, Melián CJ, Williams NM, Blüthgen N, Krasnov BR, Poulin R. 2007 Species abundance and asymmetric interaction strength in ecological networks. *Oikos* 116, 1120–1127. (doi:10.1111/j.2007.0030-1299.15828.x)

Novotny V. 2009 Beta diversity of plant-insect food webs in tropical forests: A conceptual framework. *Insect Conserv. Divers.* 2, 5–9. (doi:10.1111/j.1752-4598.2008.00035.x)

Simanonok MP, Burkle LA. 2014 Partitioning interaction turnover among alpine pollination networks: Spatial, temporal, and environmental patterns. *Ecosphere* 5, 1–17. (doi:10.1890/ES14-00323.1)

Arroyo-Correa B, Burkle LA, Emer C. 2020 Alien plants and flower visitors disrupt the seasonal dynamics of mutualistic networks. *J. Ecol.* , 1365-2745.13332. (doi:10.1111/1365-2745.13332)

Reviewer #2 (Remarks to the Author):

This manuscript studies a metaweb consisting of several multilayer networks formed by the trophic interactions of carabid beetle species on gastropod species and the herbivory interactions of the former on weeds. Specifically, the authors analyzed changes in weed seed regulation in relation to the carabid species interactions and the carabid species composition. They found that the herbivore layer of the networks mainly drives the regulation of weed seeds. Moreover, the regulation of weed seeds inversely varies with the carabid predation on the gastropods. Finally, they found that weed seed regulation was more related to link turnovers of the carabid species rather than their composition. Thus, the main claim of the study is that the ecosystem service provided by carabid species on regulating the weed seeds is governed by the behavioral plasticity of the carabid species

in being able to switch between being carnivore, herbivore or omnivore depending on the community context.

In general, this study uses a relatively novel approach (although increasingly used in ecological network studies) that analyses networks of ecosystem services by considering several layers of the networks, i.e. multilayer networks. The manuscript is generally well written, with clear arguments going straight to the points. I however have few concerns related to the way the results from the analyses are being interpreted and thus concerns on the general claim of the study.

General comments

1- The title of the study sounds very general, appealing to a general finding in any multilayer network. The study is however only on gastropod-carabid-weed networks. A more specific title would be more adapted.

2- The authors claim that carabid species adopt a behavioral plasticity in their diet. In some networks, they act as carnivores, in other networks, they act as herbivores and in others, they act as omnivores. A possibility is that these carabid species that are present in most of the networks are all omnivores. It is just that the absence of their gastropod preys make them act as herbivores, and the absence of weeds make them act as carnivores in specific networks. So, the diet choice of the carabid species is constrained only by what is available at the specific site. While a behavioral plasticity generally means that the behavioral traits of the species is expressed differently in different ecological settings, here, there is no concrete evidence that the carabid traits, i.e. their diet choice have changed in different environment. Their diet choice has just been constrained. The use of the word plasticity might then be too strong for a claim.

Minor comments

1- L67: The authors use the acronym FSE without previously mentioning what does it stand for. This acronym should have been mentioned first in L66.

2- L96: It is not clear what the word 'range' means specifically in here. Is it related to species richness? Please rephrase.

3- Figure 2: The statement that weed seed regulation increases with herbivory interaction frequency seems to contradict what is directly visible in the figure, ie, a decreasing relationship. The authors should look closely at what is actually being plotted, as this is confusing for the readers. The same comment applies to Figure S3.

4- Figure 3: (a) The sharp decrease for small values of the carnivory interaction frequency, compared to the remaining of the pattern for higher values of the frequency deserves an explanation, or at least a comment. (b) For higher number of carnivores (starting from about the value 5.5), the estimated link turnover seems to be almost uniformly high for any number of herbivores. Are these specific to any network characteristics?

5- For all figures that contain a colorbar, a colorbar title would be useful.

REVIEWERS' COMMENTS:

Reviewer #1 (Remarks to the Author):

I enjoyed reading the manuscript “Ecosystem services are governed by ecological plasticity in multilayer networks” by Gray and colleagues, in which they attempt to bring together the fields of ecological multilayer network and ecosystem services. It expands the use of ecological networks to understand the ecological mechanisms that drive the delivery of different ecosystem services. Specifically, it tries to apply the joint analysis of two important ecosystem services, weed seed and gastropod pest control, delivered by carabids through their consumption of seeds and gastropods, and how it may be affected by the distinct composition of interactions networks. This is important as we clearly more and more aware that management of ecological systems and ecosystem services can not be done in isolation, and it should be taken an approach that brings them altogether to understand their complexity and interplay.

The authors make a commendable use an extensive dataset of 374 interaction networks – the weed seed-carabids-gastropods tripartite network - obtained through from agricultural fields sampled within the Farm Scale Evaluations project, which strengthens the analysis and allows to uncover potential patterns across the networks.

They aim to test: 1) if ES of weed seed is primarily regulated at the herbivore layer (seed-carabid interactions); 2) if the weed seed control co-varies with carabids carnivore layer (predation of gastropod pest by carabids); and, finally, 3) if the seed weed regulation is affected by the turn over in the type of interactions established by carabids, i.e. switch between herbivory and carnivory.

The main conclusions are: 1) that weed control is improved (perhaps not surprisingly) by the increase in the frequency of herbivore interactions, 2) weed control is negatively affected the increase in carnivory by carabids, and 3) community changes leading to distinct availability of resources rewires the interactions by carabids between the herbivory and carnivory. Overall, the authors conclude that the ecological plasticity in the carabids networks has important consequences for ecosystem services and that multiple ecological functions need to be considered for the sustainable management of natural and agricultural systems.

Overall, the manuscript is well written and easy to read, and the authors managed to deliver the message in a clear, and concise way. Procedures for network building and statistical analysis are generally provided in sufficient detail so the reader can follow the steps taken by the authors. Bibliography seems enough to contextualize the work and discussion of the results.

There are a few points however I wish the authors to clarify and are detailed below.

Good work and good luck.

General comments:

1 - The authors put quite some emphasis on the multilayer aspect of ecological functions and ecosystems services and frame their work within that context (L.37-48). Though it is true that their system is naturally multi-layered, and should be mentioned as it is done, one aspect of the networks assembled here is that they actually miss explicit interlayer links connecting both layers. As such, what we have here is a network made up of three groups of species (weed seed – carabids - gastropods) that conform to a tripartite network (e.g. Boulay et al. 2009, Correia et al. 2019) or coupled/interdependent networks (Albrecht et al. 2014) and I think that is the term that should be used to classify the networks in this work (L.49).

As stated below, this work focus on the dynamics of the resource use by the guild that connects both layers of the system, and how that may spread across the two types of interactions (perhaps check Correia et al. 2019 or Lopez-Núñez et al 2017).

DONE. We have clarified why we chose to follow the multi-layer approach at L.54. We agree with the referee that it would be possible to conduct the analysis of the networks we have created using tripartite and coupled/interdependent network approaches, as well as multilayer network approaches. These approaches overlap in scope and are not at all exclusive. Our choice of multilayer approaches, and the associated terminology, was driven by a need to consider prey switching with some clarity. Multilayer approaches allow us to consider the individual networks (layers) in isolation and then consider their interaction as carabids move between the layers by prey-switching between weeds and molluscs. We hope and believe that this is now clear in the text.

2 – The analysis of links and interaction frequencies is largely focused on the global numbers (sum of herbivore links, sum of interaction frequencies, etc), which give a global view of the participation of the carabid guild connecting. However, the authors could go a little further by looking at the variation in the links of each species between layer across all networks. Perhaps considering some simple species-level metrics would eventually be of interest, and could be integrated in the section on herbivory-carnivory relationship (paragraph starting on L.98). For instance, carabid species degree or generality/vulnerability (both depicting the nr of partners, though in different ways. Check bipartite package's document for details and refs) for each interaction type could be obtained to calculate a herbivory:predation ratio that could be then related to the variation in gastropod/plant richness/abundance across all networks, or even with seedbank regulation. Eventually, also looking for the omnivore carabids could check if there's some tendency (e.g. using a degree-degree correlation) for the same species to have the same role across ecological functions.

DONE. At L.84, we have replaced pairwise trophic interactions with 'hypothesised, pairwise trophic links'. At L.101, we now state 'The number of hypothesised links from carabids to the weed seeds and to the gastropods in the layers of each network also co-varied inversely ($F_{1,3672} = 212.8$, $p \leq 0.001$, Supplementary Fig. S2c).'

Points 2 and 3, below, are related, we believe. It is true that our process of network construction may have both missed and over-estimated links in this system. For this reason, we chose to use the terminology 'link' for metrics of *hypothesised* link number throughout, rather than the network analysis terminology of degree. These 'link' analyses could be treated as analyses of degree and present what we would expect for the degree analyses that the referee suggests (e.g. Figure 3b).

3 – Regarding the assumptions the authors made in building the interaction networks I think they are sensible, but I would say that inevitably they are not perfect and may have some potential drawbacks. I feel that they deserve some consideration and should be presented and discussed. This can be done either where assumptions are presented or in a section of the discussion. Specifically:

3a - The assumption of diet generalisation by carabids (L.206-208) may over-estimate the existing links. Alternatively, a rarefaction approach could be used, which in turn may under-estimate the links actually occurring in the fields. Perhaps, a brief discussion of the potential effect of the assumption of a wide diet generalisation by the carabids could be added to the discussion.

DONE. We state at L.139 that 'Links were also assigned to carabids at the genus level. These processes could lead both to an under- and an over-estimate of predation on some prey.'

3b - I agree that applying this frequency weighting to links based on the product of species abundances (L.209-212) it's better than just a simple binary network. However, abundance is not the sole determinant of interaction strength, and other factors such as species preferences may result in deviations from the simple product of abundances (Vázquez et al 2005 or Staniczenko et al. 2013), which could also be affected for example by the connectance of the network. Do you think that the interaction strengths would be much affected if they were not only based on abundances? Perhaps a brief discussion of these assumption should be provided.

DONE. We state at L.214 'Future consideration of carabid-prey interaction strength may include traits, such as body size and mandible type (Bohan et al. 2011b).'

4 – Finally, the authors seem to imply the existence of a trade-off in managing the two types of organisms (L.156-158) not desired at crop fields (weeds and gastropods): to reduce the use of herbicides one should focus on reducing the potential to prey-switching towards gastropods (due to the trophic plasticity by carabids), which may require an extra input of molluscicides. In that case, we are just moving from one chemical to another. Does it mean that it wouldn't be possible to make a joint reduction of the two types of external inputs? Does it suggest that such inputs are inevitable, and we are just faced with a decision about the ration of those inputs?

DONE. At L.159 we have added the clause ', and whether it would result or not in pesticide reduction,' to indicate that the actual effect of our argument is unknown, and might decrease, increase or leave unchanged the amounts of pesticides used and their balance.

Detailed comments to the authors:

L.65-66: After Farm Scale Evaluations add "FSE".

DONE.

L.73: Quite not understand this. The nodes of the network are the actual species and not their abundances. Please, clarify or/and remove "abundances".

DONE. We have removed the term abundance.

Section "Results and discussion": I would suggest dividing this section according to the three hypothesis state above to make it easier to follow. Unless, not allowed by the formatting rules of the journal.

NO CHANGE MADE. The structure that is used is considered appropriate to the journal.

1st paragraph: It would be interesting, as a general overview, to have a summary of the interactions (total and for each type) and species across all networks (mean and range).

DONE. We found that a table summarising the numbers of interactions and their ranges was either too complicated, and better represented as a graphic, or too simple leading to further questions about the tabulated values for different levels of interaction. In answer to a comment below, we have promoted Figure S4 to the main text (as Figure 3b) and believe that this new figure serves for this purpose.

L.83: Aggregated networks usually means that all layers are merged to form a single monolayer network that sums up all the nodes and links of the multilayer network (Kivelä et al 2014, pp. 228), which is not the case here, and it's not synonymous of metaweb. Better remove "aggregated network" here and call it "tripartite metaweb".

DONE. We changed the term metaweb to 'composite network' throughout, denoting a composite of all layers across all sites.

L.102: But Fig2c reports the number of herbivores vs number of predators (which could read as number of nodes), rather than the number of links of each type established by carabids. Please clarify.

DONE. At L.101, we now state 'The number of hypothesised links from carabids to the weed seeds and to the gastropods in the layers of each network also co-varied inversely ($F_{1,3672} = 212.8$, $p \leq 0.001$, Supplementary Fig. S2c).'

L107: I think this plot is worth being as main figure, as it helps showing the general separation between the two functions provided by different species of carabids, and that omnivore species are not that efficient at delivering both functions. Perhaps as an extra panel of fig3.

DONE. Figure moved to panel b) of Figure 3.

L.110-114: It can be noticed on both figure 3b and figure S5 an apparent change in turnover once the number of carnivore species reaches a certain number of species (transition from 5 to 6 species). This seems to be ignored, but I wonder if there's some reason/meaning for that.

DONE. We now state at L.122 that 'Higher dissimilarities at predator link values of 5 and above may reflect carabid predation preferences for molluscs, which remain to be tested.'

L.116: The figure on the left could be a nice visualization and I like how it looks, but I don't think it works very well here. For example, the first column of networks (no links to gastropods) should have no red links but they are present. Similar thing happens on the rows. It may be a matter of size or space available. Also, isn't it showing the same as fig S4? Consider removing or redrawing. If figure is kept add a) and b) to the respective panels.

DONE. Graphic modified to reflect increasing numbers of herbivore or predation interactions. Graphic moved to Figure 1. Here, the graphic serves to indicate the variation in structure observed across the composite network using selected examples of the individual networks. This is similar to but distinct from Figure S4.

L.141: Close parenthesis missing after references.

DONE.

L.169: Perhaps add a scheme showing the split-field design used.

NOT DONE. These schemes are extensively presented in the original papers from the FSE, which are readily available. Papers subsequent to these (for example Ma et al 2019 doi: 10.1038/s41559-018-0757-2) cite these original papers and we do not believe reproducing the text for this large design is warranted here.

L.171-172: I understand that all the details are already published in previous works and there's no need (or the space) to give them again here. However, scale at which the data were collected (plot size), distance between fields, or an idea of how independent the half-fields are should be provided, I think.

DONE. We now state 'The great majority of fields were below 15 Ha in size, and typically varied between 2 and 10 Ha depending upon the crop (Bohan et al. 2005; Champion et al. 2003).'

L.183: How many soil samples from each field were collected?

DONE. We now state 'The weed seedbank was measured by taking 8 soil samples in each field, at 2 and 32 m points on four transects, prior to the crops being sown and after harvest (Heard *et al.* 2003).'

L.187-188: How many seed traps?

DONE. We now state 'Eight seed rain traps, placed at 2 and 32 m along 4 of the transects, sampled the rain of weed seed throughout the growing season (Heard *et al.* 2003).'

L.193-194: Given that seedbank total of weed species was obtained as a before-after, does the relative abundance refers only to animals (gastropods and carabids)? If that is so make explicit. If for weeds the relative abundance refers to the before-after number of seeds, perhaps describe it in a separate sentence. Otherwise, seems a bit confusing as it is. Moreover, before and after what? Reckon, as suggested further down in L. 224-225, that it refers to harvest of crops at each field. It is not clear. Also, is this the difference between before and after, or the sum of counts in all samples before and after. If the later, why are you summing the counts from both instances? Clarify, please.

DONE. Text modified at L.191-196 and at L.225-227 to clarify the process of treating the data for analysis.

L.202: Individual? Species?

DONE. Text changed to species.

L203-206: Isn't this saying the same as the previous sentence? Seems redundant.

NO CHANGE MADE. This text is not redundant because it indicates the method used for assigning links to the resources. While similar to the assignment of links to the carabids, these processes are distinct and should be explicitly stated.

L.224: If I correctly understand, any deviation from unit (i.e. no seed difference between t1 and t0) would be due to carabids regulating the seedbank, right? Isn't this a too strong assumption? This would imply that any other factors (ant predation, wind, or other types of dispersal, and factors) are mostly negligible.

NO CHANGE MADE. As we note at L.49, previous work has demonstrated that carabids have been shown to regulate the weed seedbank. We therefore limit our work in this paper to tests of the effects of carabids on weed seed regulation. We do not attempt in this paper to account for other explanations of seedbank change including herbicides and the biotic effects of ants, birds and mammals, for example. While we do not discuss this here, other sources of management are important for explaining weed seedbank abundance and variation (see, for example, Bohan et al. (2011) doi:10.1111/j.1365-3180.2011.00860.x).

L.240: But it seems that the decomposition of link dissimilarity was not actually done here. In the paper cited at the end of this sentence (Poisot et al 2012) it is proposed an approach to perform that decomposition, and you could present not only species and interaction turnover, but also how much of the interaction turnover is due to changes in species composition and to the same species establishing distinct interactions - rewiring. Perhaps that approach could be employed here as well. In a more detailed approach, interaction turnover due to species turnover could be further decomposed into the contribution prey turnover, carabid turnover and simultaneous turnover of plant and carabids (Novotny 2009, Simanonok & Burkle, 2014, Arroyo-Correa et al. 2020).

DONE. We removed the sentence at L.239-240 that cites Poisot et al. (2012). The network construction that we have used hypothesises links to be present whenever the appropriate nodes were present, and this prevents the method of Poisot et al. (2012) from being used here.

L.252, 260, 265, 269: All these models are mixed models, so perhaps write it explicitly. Moreover, all of them seem to use Gaussian error distribution, given the transformation applied to data, except in the case of the analysis of link richness. In this later case, which error distribution was used? If in all cases Gaussian error distribution, perhaps simply call the models Linear Mixed Models.

DONE. All models were Generalised linear mixed effects model that used the Gaussian distribution. Text was changed in numerous places in the Statistical analysis section to reflect this.

Figure 1: I'm not entirely convinced that this particular display of the network is very effective, given the high number of links involved. Perhaps a simpler display, with weeds at the bottom, carabids in the middle and gastropods at the top, spread across the width of the page would allow a better view of the profusion on links in the metaweb.

NO CHANGE MADE. We chose this Hive plot because it was the simplest method that we have found to display something of the complex structure and variation in the composite web.

Figure 2: Add to the legend what is the meaning of the lines, and how to interpret the dashed line. Same for Figure S3 and S4.

DONE.

Figure 3: Add next to the colour scale on the right that it refers to dissimilarity gradient. Same for Figure S5.

DONE.

Figure S2: Perhaps organize the different plot as in the Figure S3.

DONE.

Suggested bibliography (cite, or not, at your discretion):

Boulay R, Carro F, Soriguer RC, Cerdá X. 2009 Small-scale indirect effects determine the outcome of a tripartite plant-disperser-granivore interaction. *Oecologia* 161, 529–37. (doi:10.1007/s00442-009-1404-z)

Correia M, Rodríguez-Echeverría S, Timóteo S, Freitas H, Heleno R. 2019 Integrating plant species contribution to mycorrhizal and seed dispersal mutualistic networks. *Biol. Lett.* 15, 20180770. (doi:10.1098/rsbl.2018.0770)

Albrecht J, Gertrud Berens D, Jaroszewicz B, Selva N, Brandl R, Farwig N. 2014 Correlated loss of ecosystem services in coupled mutualistic networks. *Nat. Commun.* 5, 3810. (doi:10.1038/ncomms4810)

Kivelä M, Arenas A, Barthelemy M, Gleeson JP, Moreno Y, Porter MA. 2014 Multilayer networks. *J. Complex Networks* 2, 203–271. (doi:10.1093/comnet/cnu016)

Vázquez DP, Melián CJ, Williams NM, Blüthgen N, Krasnov BR, Poulin R. 2007 Species abundance and asymmetric interaction strength in ecological networks. *Oikos* 116, 1120–1127. (doi:10.1111/j.2007.0030-1299.15828.x)

Novotny V. 2009 Beta diversity of plant-insect food webs in tropical forests: A conceptual framework. *Insect Conserv. Divers.* 2, 5–9. (doi:10.1111/j.1752-4598.2008.00035.x)

Simanonok MP, Burkle LA. 2014 Partitioning interaction turnover among alpine pollination networks: Spatial, temporal, and environmental patterns. *Ecosphere* 5, 1–17. (doi:10.1890/ES14-00323.1)

Arroyo-Correa B, Burkle LA, Emer C. 2020 Alien plants and flower visitors disrupt the seasonal dynamics of mutualistic networks. *J. Ecol.* , 1365-2745.13332. (doi:10.1111/1365-2745.13332)

We have cited:

Boulay R, Carro F, Soriguer RC, Cerdá X. 2009 Small-scale indirect effects determine the outcome of a tripartite plant-disperser-granivore interaction. *Oecologia* 161, 529–37. (doi:10.1007/s00442-009-1404-z)

Correia M, Rodríguez-Echeverría S, Timóteo S, Freitas H, Heleno R. 2019 Integrating plant species contribution to mycorrhizal and seed dispersal mutualistic networks. *Biol. Lett.* 15, 20180770. (doi:10.1098/rsbl.2018.0770)

Albrecht J, Gertrud Berens D, Jaroszewicz B, Selva N, Brandl R, Farwig N. 2014 Correlated loss of ecosystem services in coupled mutualistic networks. *Nat. Commun.* 5, 3810. (doi:10.1038/ncomms4810)

Reviewer #2 (Remarks to the Author):

This manuscript studies a metaweb consisting of several multilayer networks formed by the trophic interactions of carabid beetle species on gastropod species and the herbivory interactions of the former on weeds. Specifically, the authors analyzed changes in weed seed regulation in relation to the carabid species interactions and the carabid species composition. They found that the herbivore layer of the networks mainly drives the regulation of weed seeds. Moreover, the regulation of weed seeds inversely varies with the carabid predation on the gastropods. Finally, they found that weed seed regulation was more related to link turnovers of the carabid species rather than their composition. Thus, the main claim of the study is that the ecosystem service provided by carabid species on regulating the weed seeds is governed by the behavioral plasticity of the carabid species in being able to switch between being carnivore, herbivore or omnivore depending on the community context.

In general, this study uses a relatively novel approach (although increasingly used in ecological network studies) that analyses networks of ecosystem services by considering several layers of the networks, i.e. multilayer networks. The manuscript is generally well written, with clear arguments going straight to the points. I however have few concerns related to the way the results from the analyses are being interpreted and thus concerns on the general claim of the study.

General comments

1- The title of the study sounds very general, appealing to a general finding in any multilayer network. The study is however only on gastropod-carabid-weed networks. A more specific title would be more adapted.

NO CHANGE MADE. We believe that our findings are general, both for this important system that is present across much of European arable farmland and for other systems. Our title serves the purpose of being both a generic description of our work and a hypothesis for future work to demonstrate the utility of understanding the interactions between ecosystem services in terms of ecological networks. We have modified the title in line with the editor's suggestion 'Ecological plasticity governs ecosystem services in multilayer networks'.

2- The authors claim that carabid species adopt a behavioral plasticity in their diet. In some networks, they act as carnivores, in other networks, they act as herbivores and in others, they act as omnivores. A possibility is that these carabid species that are present in most of the networks are all omnivores. It is just that the absence of their gastropod preys make them act as herbivores, and the absence of weeds make them act as carnivores in specific networks. So, the diet choice of the carabid species is constrained only by what is available at the specific site. While a behavioral plasticity generally means that the behavioral traits of the species is expressed differently in different ecological settings, here, there is no concrete evidence that the carabid traits, i.e. their diet choice have changed in different environment. Their diet choice has just been constrained. The use of the word plasticity might then be too strong for a claim.

NO CHANGE MADE. We note at L.45 that carabid communities contain species from specialist granivores to more generalist predators and these classifications are habitually used in carabidology, even as this thinking is undergoing revision (see L.144). Our work simply suggests that these definitions, which reflect autecological considerations and not behavioural plasticity, are too categorical (see L.144 - 155). Our point is that carabids have *potential* plasticity and, then, their *realised* food intake is constrained by their food item context. *Pterostichus melanarius* has typically been considered a generalist carnivore. More recently, it has begun to be treated as an omnivore (see Bohan et al. 2011a). We show here that it can also be an herbivore; a classification that would not have been considered 10 years ago.

Minor comments

1- L67: The authors use the acronym FSE without previously mentioning what does it stand for. This acronym should have been mentioned first in L66.

DONE.

2- L96: It is not clear what the word 'range' means specifically in here. Is it related to species richness? Please rephrase.

DONE. We have changed the term range to 'count' for clarity.

3- Figure 2: The statement that weed seed regulation increases with herbivory interaction frequency seems to contradict what is directly visible in the figure, ie, a decreasing relationship. The authors

should look closely at what is actually being plotted, as this is confusing for the readers. The same comment applies to Figure S3.

DONE. We now state at L.229 that “Negative values of this metric indicate a decline in the size of the weed seedbank from t_0 to t_1 . Following Bohan *et al.* (2011a), negative relationships between the metric of regulation and carabid counts are indicative of seedbank regulation by these beetles.”

4- Figure 3: (a) The sharp decrease for small values of the carnivory interaction frequency, compared to the remaining of the pattern for higher values of the frequency deserves an explanation, or at least a comment. (b) For higher number of carnivores (starting from about the value 5.5), the estimated link turnover seems to be almost uniformly high for any number of herbivores. Are these specific to any network characteristics?

DONE. We now state at L.122 that ‘Higher dissimilarities at predator link values of 5 and above may reflect potential carabid predation preferences for molluscs, which remain to be tested.’

5- For all figures that contain a colorbar, a colorbar title would be useful.

DONE.